# Synthesis of Spin-Labelled Bergamottin: A Potent CYP3A4 Inhibitor with Antiproliferative Activity

**DOI:** 10.3390/ijms21020508

**Published:** 2020-01-13

**Authors:** Balázs Zoltán Zsidó, Mária Balog, Nikolett Erős, Miklós Poór, Violetta Mohos, Eszter Fliszár-Nyúl, Csaba Hetényi, Masaki Nagane, Kálmán Hideg, Tamás Kálai, Balázs Bognár

**Affiliations:** 1Department of Pharmacology and Pharmacotherapy, University of Pécs, Medical School, Szigeti út 12, H-7624 Pécs, Hungarycsabahete@yahoo.com (C.H.); 2Institute of Organic and Medicinal Chemistry, University of Pécs, Medical School, Honvéd utca 1, H-7624 Pécs, Hungary; maria.balog@aok.pte.hu (M.B.); erosniki93@gmail.com (N.E.); kalman.hideg@aok.pte.hu (K.H.); tamas.kalai@aok.pte.hu (T.K.); 3Department of Pharmacology, University of Pécs, Faculty of Pharmacy, Szigeti út 12, H-7624 Pécs, Hungary; poor.miklos@pte.hu (M.P.); mohos.violetta@gytk.pte.hu (V.M.); eszter.nyul@aok.pte.hu (E.F.-N.); 4János Szentágothai Research Center, University of Pécs, Ifjúság útja 20, H-7624 Pécs, Hungary; 5Department of Biochemistry, School of Veterinary Medicine, Azabu University, 1-17-71 Fuchinobe, Chuo-ku, Sagamihara, Kanagawa 252-5201, Japan; nagane@azabu-u.ac.jp

**Keywords:** bergamottin, nitroxide, CYP3A4 inhibition, anticancer activity

## Abstract

Bergamottin (BM, **1**), a component of grapefruit juice, acts as an inhibitor of some isoforms of the cytochrome P450 (CYP) enzyme, particularly CYP3A4. Herein, a new bergamottin containing a nitroxide moiety (SL-bergamottin, SL-BM, **10**) was synthesized; chemically characterized, evaluated as a potential inhibitor of the CYP2C19, CYP3A4, and CYP2C9 enzymes; and compared to BM and known inhibitors such as ketoconazole (KET) (3A4), warfarin (WAR) (2C9), and ticlopidine (TIC) (2C19). The antitumor activity of the new SL-bergamottin was also investigated. Among the compounds studied, BM showed the strongest inhibition of the CYP2C9 and 2C19 enzymes. SL-BM is a more potent inhibitor of CYP3A4 than the parent compound; this finding was also supported by docking studies, suggesting that the binding positions of BM and SL-BM to the active site of CYP3A4 are very similar, but that SL-BM had a better ∆G_bind_ value than that of BM. The nitroxide moiety markedly increased the antitumor activity of BM toward HeLa cells and marginally increased its toxicity toward a normal cell line. In conclusion, modification of the geranyl sidechain of BM can result in new CYP3A4 enzyme inhibitors with strong antitumor effects.

## 1. Introduction

Bergamottin (i.e., 5-geranoxypsoralen (**1**)) is a natural furanocoumarin that was originally detected in bergamot oil (*Citrus bergamia*) [1] and is mainly responsible, together with 6′,7′-dihydroxybergamottin (DHB, **2**), for “grapefruit juice/drug” interactions (Figure 1). Grapefruit juice has been found to cause a marked increase in the oral bioavailability of many therapeutic agents (e.g., dihydropyridines [2], ethinylestradiol [3], midazolam [4], cyclosporine A [5], and lovastatin [6]) primarily by inhibiting the CYP3A4 enzyme. Several mechanisms have been reported regarding the CYP3A4 inhibitory effect of bergamottin such as decreased protein expression or the reversible inhibition of the enzyme [7,8,9,10].

Row et al. studied the inhibitory effects of a series of furanocoumarin analogues on the CYP3A4 enzyme. They found that the furan ring and the alkyloxy group were necessary for inhibition, and hydrophilic groups at 6′,7′-positions enhanced the potency compared to the alkenyl group [11]. Although bergamottin is a less potent CYP3A4 inhibitor than DHB [12], it is a stronger inhibitor of some other CYP subfamilies such as CYP1A and 2B [13]. Bergamottin was also found to inhibit CYP2A6, 2C9, 2C19, 2D6, and 2E1 enzymes in human liver microsomes [14].

Bergamottin appeared to be a potential anticancer agent against various tumor cell lines by the regulation of several cancer-related pathways [15]. Through the inhibition of STAT3 activation, BM inhibited the proliferation of breast cancer cells and multiple myeloma [16]. BM attenuated cell migration and invasion of fibrosarcoma and lung cancer cell lines [17,18]. In combination with simvastatin, BM suppressed the TNF-*α*-induced anti-proliferative and pro-apoptotic processes of KBM-5 myeloid leukemia cells [19].

The unpaired electrons of stable nitroxide free radicals allow the nitroxides to take part in one-electron oxidation and reduction processes, which make nitroxides potent unnatural antioxidants [20]. As a consequence, nitroxides exert a cytoprotective action against oxidative stress induced by cytotoxic drugs or pathological processes, such as ischemia-reperfusion and inflammation [21,22,23]. However, some piperidine types of nitroxides also exert cytotoxic and pro-oxidant effects, particularly on cancer cells [24,25].

We have reported in several studies that the modification of biomolecules (caffeic acid phenethyl ester, resveratrol, and curcumin) with nitroxides for use as “in statu nascendi acting” antioxidants may be beneficial on their activity. For instance, better antioxidant, antiproliferative, anti-inflammatory and cell-protective compounds can be achieved by these modifications [26,27,28,29,30,31].

Herein, we report the synthesis, CYP (2C9, 2C19, and 3A4) enzyme inhibition, and the anticancer activity of a new paramagnetic bergamottin analog compared to the parent compound and known CYP inhibitors. Furthermore, CYP3A4 enzyme inhibition was also evaluated by docking studies.

## 2. Results

### 2.1. Synthesis of Spin-Labelled Geraniol (**7**)

For the synthesis of nitroxide-modified geraniol (**7**), the first step was to alkylate the ethyl acetoacetate with paramagnetic allylic bromide (**3**) [32] in acetone and in the presence of K_2_CO_3_ at ambient temperature to obtain compound **4** with 67% yield. The ketoester was hydrolyzed with a NaOH/ethanol solution and then acidified with aq. H_2_SO_4_. The elimination of CO_2_ happened spontaneously to yield ketone **5**, in which the Horner–Wadsworth–Emmons reaction with triethyl phosphonoacetate led to the formation of paramagnetic Z/E esters (**6**) at a 1:4 ratio. After purification, the reduction of *E*-ester **6** with sodium-bis (2-methoxyethoxy) aluminum hydride (SMEAH) in anhydrous toluene afforded *E*-4-(6-hydroxy-4-methyl-4-hexene-1-ylidene)-2,2,6,6-tetramethyl-piperidine-1-oxyl (**7**, spin-labelled geraniol) (Scheme 1). 

### 2.2. Synthesis of Spin-Labelled Bergamottin (**10**)

The alkylation of 4-hydroxy-7*H*-furo[3,2-g]chromen-7-one (bergaptol, **9**) may result in new bergamottin derivatives. To achieve a nitroxide ring motif containing bergamottin (**10**), paramagnetic geranyl bromide (**8**) was synthesized via the treatment of **7** with methanesulfonyl chloride and followed by LiBr. Bergaptol (**9**) was then alkylated with compound **8** in dry acetone in the presence of K_2_CO_3_ and catalytic amount of sodium iodide to afford SL-BM (**10**) (Scheme 2).

### 2.3. Inhibition of the CYP Enzymes by BM and SL-BM

BM proved to be a strong inhibitor of each CYP enzyme tested, while SL-BM showed considerable inhibitory effects toward the CYP2C19 and CYP3A4 enzymes (Figure 2). BM was a two-fold stronger inhibitor of CYP2C9 than the positive control warfarin (Table 1); the inhibitory effect induced by SL-BM was also statistically significant, but only a slight decrease in metabolite formation resulted (even at a four-fold concentration vs. the substrate). SL-BM was approximately two times weaker, while BM induced almost a seven-times stronger inhibitory effect toward CYP2C19 relative to the inhibitory effect of the positive control ticlopidine. Despite the fact that BM proved to be a considerably stronger inhibitor of both the CYP2C enzymes tested, SL-BM showed a significantly stronger inhibitory action toward CYP3A4 (Figure 2). BM and SL-BM were 1.7 and 8.5 times weaker inhibitors of CYP3A4 than the positive control ketoconazole (Table 1).

### 2.4. Modeling Studies

Ligands KET, BM, and **10** were docked to the binding pocket of CYP3A4, which is located above the heme ring. The docking of KET tested the applicability of the methodology for producing a close to crystallographic bound KET conformation. In the cases of BM and SL-BM, the binding modes were de novo, as described in the present study.

Binding mode of KET. According to the holo CYP3A4 crystallographic structure (Protein Data Bank (PDB) code 2v0m), KET (Table 2) is coordinated to the iron of the heme with a nitrogen of the imidazole ring, and the distance between the N atom and Fe^3+^ is 2.7 Å. Interacting amino acid residues are listed in Table 2. The heme-bound crystallographic ligand conformation of KET was used to verify the applicability of our computational docking protocol (see Methods for details) for the atomic resolution calculation of the KET binding mode. The docking of KET into the ligand-free binding pocket of the 2v0m structure was successful, and the crystallographic ligand binding mode of KET was reproduced at an RMSD value of 2.3 Å (Figure 3) in the top 2nd rank. The main interactions were reproduced. That is, the phenyl rings of F304 and KET were parallel, forming *π–π* interactions, and A370 formed a hydrophobic interaction with the methyl group of KET. The positively charged sidechain of R372 interacted with the partial negative charge of the oxo group of KET.

Binding modes of BM and SL-BM. To determine the binding modes of BM and SL-BM, the same docking protocol was applied as that used in the case of KET, as described in the previous paragraph. In both cases, the top 1st rank binding mode was very similar to that of KET (Figure 4). The furo[3,2-g]chromen-7-one ring was located above the heme occupying the subpockets of the imidazole ring of KET. The furan oxygen was coordinated to the heme for both ligands. The O–Fe^3+^ distances were 4.0 and 4.2 Å for compounds **10** and **1**, respectively (Figure 5). In the bound position, the furo[3,2-g]chromen-7-one ring of the ligands occupied the same subpocket and their side-chains were also in a similar position. The furo[3,2-g]chromen-7-one ring of the ligands was parallel to the heme, possibly forming π-stacking interactions, which was similar to KET (as described in the previous paragraph). Eight and seven amino acid residues were found to interact (Methods) with SL-BM and BM, respectively; of which five residues were common. Compounds **10** and **1** both interacted with CYP3A4 through H-bonds between the hydrophilic amino acid (T309) and the furan oxygen. Hydrophobic interactions occurred between the M114, F241, I301, and F304 amino acids and the methyl groups of the ligands. F304 played a role in the binding of each (KET, BM, and SL-BM) ligand (Table 2).

### 2.5. Cytotoxicity of BM and SL-BM

The cytotoxicity of BM and SL-BM was analyzed by a WST Cell Viability & Proliferation assay performed using NIH3T3 (murine embryonic fibroblast) and HeLa (human cervix carcinoma) cells. Neither bergamottin nor SL-bergamottin showed a significant toxicity toward fibroblasts (IC_50_ > 50 µM). BM did not decrease the viability of the HeLa cells (IC_50_ > 50 µM); however, SL-BM induced a significant loss in the viability of the HeLa cell line (IC_50_ = 17.32 µM). These results demonstrate the selective toxicity of SL-BM toward cancer cells (Figure 6).

## 3. Discussion

A new nitroxide moiety containing bergamottin analog (**10**) has been synthesized and evaluated for use as an inhibitor of CYP (2C9, 2C19, and 3A4) enzymes and compared to bergamottin (**1**) and known inhibitors of these enzymes. The cytotoxicity toward cancer and noncancer cell lines was also investigated.

BM induced a 50% inhibition of the metabolite formation at 0.2- and 0.4-fold concentrations vs. the substrates in the CYP2C19 and CYP3A4 assays, respectively (Table 1). The IC_50_ values of BM toward these enzymes were in the low micromolar range, which agrees well with the previously reported data [14,34,35,36]. Furthermore, BM proved to also be an inhibitor of CYP2C9, showing 50% inhibition of metabolite formation at approximately a three-fold concentration vs. the substrate. Previous studies also reported the significant inhibitory effect of BM on CYP2C9 enzymes [11,14,35,36,37]. As our results demonstrated, SL-BM only slightly inhibits CYP2C9 and is almost a 15-fold weaker inhibitor of CYP2C19 than BM (Table 1). However, SL-BM was a five-fold stronger inhibitor of CYP3A4 compared to BM, showing a strong inhibitory efficacy similar to that of the positive control ketoconazole. The enhanced inhibitory activity of SL-BM compared to that of BM was also supported by docking experiments, where the binding of SL-BM was more favorable than that of BM (∆G_bind_(−10.4 vs. −9.2 kcal/mol)). The difference in the inhibitory activities of SL-BM and BM may be attributed to the H-acceptor property of the nitroxide, as it was suggested by Row et al. [11].

BM and SL-BM seemed to be nontoxic to normal cells since they did not significantly decrease the viability of NIH3T3 fibroblasts in our toxicity assay. As far as we know, this is the first report about the anticancer activity of bergamottin toward HeLa cells. As shown in previous reports, although BM showed an inhibition effect on many cancer cell lines, such as HT-1080 fibrosarcoma [17], U266 multiple myeloma [18], HepG2 liver cancer, BGC-823 gastric cancer, HL-60 promyelotic leukemia [38], and A549 lung cancer cells [16], we did not observe BM to be significantly cytotoxic toward the HeLa cell line. Nevertheless, the insertion of a nitroxide moiety (**10**, IC_50._ = 17.32 µM) resulted in the cancer-specific cytotoxic activity of the parent compound (**1**, IC_50_ > 50 µM). Therefore, compound **10** may be a good starting point for the development of new CYP3A4 enzyme inhibitors with elevated anti-proliferative effects.

## 4. Materials and Methods

### 4.1. Chemistry

#### 4.1.1. General

The mass spectra were recorded with a Thermoquest Automass Multi system (ThermoQuest, CE, Instruments, Milan, Italy) operated in EI mode (70 eV). Elemental analyses were carried out with a Fisons EA 1110 CHNS elemental analyzer (Fisons Instruments, Milan, Italy) The melting points were determined with a Boetius micro-melting point apparatus (Franz Küstner Nachf. K. G., Dresden, Germany). The ^1^H NMR spectra were recorded with a Bruker Avance 3 Ascend 500 system (Bruker BioSpin Corp., Karsluhe, Germany) operated at 500 MHz, and the ^13^C NMR spectra were obtained at 125 MHz in CDCl_3_ or DMSO-d_6_ at 298 K. The “in situ” reduction of the nitroxides was achieved by the addition of five equivalents of hydrazobenzene (DPPH/radical). The IR spectra were obtained with a Bruker Alpha FT-IR instrument (Bruker Optics, Ettlingen, Germany) with an ATR support on a ZnSe plate. Flash column chromatography was performed on Merck Kieselgel 60 (0.040–0.063 mm). Qualitative TLC was carried out on commercially available plates (20 cm × 20 cm × 0.02 cm) coated with Merck Kieselgel (Darmstadt, Germany) GF_254_. Compound 3 [32], was synthesized as previously described. All the other reagents were purchased from Sigma Aldrich (St. Louis, MO, USA), Molar Chemicals (Halásztelek, Hungary) or TCI (Tokyo, Japan). ^1^H-NMR and ^13^C-NMR spectra of new compounds are available as supporting data (see Appendix A).

#### 4.1.2. Preparation of Compounds

*Ethyl 2-acetyl-4-(1-oxyl-2,2,6,6-tetramethyl-piperidin-4-ylidene)butanoate (4)*: Potassium carbonate (11.09 g, 80.0 mmol), ethyl acetoacetate (6.50 g, 50.0 mmol), and 18-crown-6 (10 mg) were dissolved in anhydrous acetone (30 mL), allylic bromide (3) (2.61 g, 10.0 mmol) dissolved in dry acetone (10 mL) was added dropwise, and the mixture was stirred and refluxed for 24 h. The solvent was evaporated, the residue was partitioned between H_2_O (25 mL) and CH_2_Cl_2_ (30 mL), after separation of the phases, the water phase was extracted with further CH_2_Cl_2_ (2 × 20 mL). The combined organic phases were dried over MgSO_4_, filtered, evaporated, and the residue was purified by column chromatography (hexane-Et_2_O) to achieve compound 4 2.07 g (67%) as a red oil. R_f_: 0.2 (hexane-Et_2_O, 2:1), MS m/z (%): 310 (M^+^,42), 166 (13), 135 (100), 107 (56). Anal. calcd. for C_17_H_28_NO_4_. C 65.78; H 9.09; N 4.51, Found: C 65.55, H 9.36, N 4.42. IR (neat): $\bar{\nu }\;$= 1738, 1714 cm ^−1^. ^1^H NMR (500 MHz, CDCl_3_ + (PhNH)_2_): *δ* = 5.21 (t, 1H, *J* = 7.3 Hz), 4.26 (q, 2H, *J* = 7.1 Hz), 3.53 (t, 1H, *J* = 7.5 Hz), 2.67 (dt, 2H, *J* = 7.3, 2.1 Hz), 2.29 (s, 3H), 2.23 (s, 2H), 2.12 (s, 2H), 1.33 (t, 3H, *J* = 7.1 Hz), 1.20, 1.17 (2s, 12H). ^13^C NMR (125 MHz, CDCl_3_ + (PhNH)_2_): *δ* = 202.7, 169.5, 135.5, 121.1, 61.5, 60.2, 59.9, 49.5, 41.6, 29.2, 26.4, 14.2.

*5-(1-Oxyl-2,2,6,6-tetramethylpiperidin-4-ylidene)pentan-2-one (5)*: To the ethanolic (20 mL) solution of ketoester (4) (1.55 g, 5.0 mmol), 10% aqueous NaOH (20 mL) was added, and the mixture was refluxed for 30 min. After additional 24 h stirring at room temperature, the solvent was evaporated off and the residue was acidified with 5% H_2_SO_4_ and extracted with EtOAc (3 × 20 mL). The combined organic phases were dried (MgSO_4_), filtered, evaporated, and purified by flash column chromatography (hexane-Et_2_O) to yield compound 5 as a red oil, 702 mg (59%); R_f_: 0.23 (hexane-Et_2_O, 2:1). MS (EI): *m*/z (%) = 238 (M^+^, 41), 224 (100), 180 (16), 166 (60), 43 (100). Anal. calcd. for C_14_H_24_NO_2_: C, 70.55; H, 10.15; N, 5.88. Found: C, 70.39; H, 10.21; N, 5.77. IR (neat): $\bar{\nu }$ = 1714, 1633 cm^−1^. ^1^H NMR (500 MHz, CDCl_3_ + (PhNH)_2_): *δ* = 5.23 (t, 1H, *J* = 7.2 Hz), 2.50 (t, 2H, *J* = 7.5 Hz), 2.35 (q, 2H, *J* = 7.4 Hz), 2.20 (s, 2H), 2.16 (s, 3H), 2.11 (s, 2H), 1.19, 1.17 (2s, 12H). ^13^C NMR (125 MHz, CDCl_3_ + (PhNH)_2_): *δ* = 208.4, 133.4, 124.0, 60.3, 60.2, 49.4, 43.9, 41.6, 30.0, 22.0.

*(E)-Ethyl 6-(1-oxyl-2,2,6,6-tetramethylpiperidin-4-ylidene)-3-methylhex-2-enoate (6)*: To a stirred suspension of NaH (96 mg, 4.0 mmol) in dry toluene (15 mL), triethyl phosphonoacetate (1.12 g, 5.0 mmol) was added at 0 °C under a N_2_ atmosphere. After stirring for 10 min, 5 ketone (720 mg, 3.0 mmol) was added, and the mixture was stirred and refluxed for another 30 min. After cooling to room temperature, brine (20 cm^3^) and ether (20 cm^3^) were added, the organic phase was separated, and the aqueous phase was washed with ether (2 × 20 cm^3^). Then, the combined organic phase was dried (MgSO_4_), filtered and evaporated, and the residue was purified by flash column chromatography (hexane-Et_2_O) to yield 315 mg (34%) of compound 6 as a reddish-brown oil. R_f_: 0.40 (hexane-Et_2_O, 2:1); MS m/z (%) 308 (M^+^, 38), 278 (6), 166 (11), 74 (100). Anal. calcd. for C_18_H_30_NO_3_: C, 70.09; H, 9.80; N, 4.54. Found: C, 70.19; H, 9.91; N, 4.60. IR (neat): $\bar{\nu }$ = 1713, 1648 cm ^−1^. ^1^H NMR (500 MHz, CDCl_3_ + (PhNH)_2_): *δ* = 5.78 (s,1H), 5.25 (t, 1H, *J* = 6.3 Hz), 4.24 (q, 2H, *J* = 7.1 Hz), 2.30–2.24 (m, 5H), 2.21 (s, 2H), 2.14 (s, 2H), 1.36 (t, 3H*, J* = 7.1 Hz), 1.21 (s, 6H), 1.19 (s, 6H). ^13^C NMR (125 MHz, CDCl_3_ + (PhNH)_2_): *δ* = 166.8, 159.3, 133.2, 124.4, 116.0, 60.3, 60.2, 59.5, 49.5, 41.7, 41.2, 25.7, 18.9, 14.4.

*(E)-4-(6-hydroxy-4-methylhex-4-en-1-ylidene)-2,2,6,6-tetramethylpiperidin-1-yloxy (7)*: To a stirred solution of ester (6) (716 mg, 2.0 mmol) in abs. toluene (20 mL), SMEAH (1.21 g, 6.0 mmol) dissolved in toluene (10 mL) was added dropwise under N_2_ at −30 °C. Then, the mixture temperature was allowed to rise to room temperature, and the reaction was stirred for a further 30 min. After completion of the reaction, the solution was added in portions to the mixture of 10 % NaOH (25 mL) and crushed ice (50 g). The phases were separated, and the aqueous phase was extracted with a mixture of THF/Et_2_O (1:4) (3 × 20 mL). The combined organic phases were dried (MgSO_4_), filtered, and evaporated. The title compound (7) was obtained after column chromatography (hexane-EtOAc) as a red oil (440 mg, 83%). R_f_: 0.23 (hexane/EtOAc, 2:1), MS m/z (%) 266 (M^+^, 43), 95 (51), 74 (100), Anal. calcd. for C_16_H_28_NO_2_: C, 72.14; H, 10.59; N, 5.26. Found: C, 72.31; H, 10.51; N, 5.37. IR (neat): $\bar{\nu }$ = 3412, 1670, 1098 cm ^−1^. ^1^H NMR (500 MHz, CDCl_3_ + (PhNH)_2_): *δ* = 5.48 (dt,1H, *J* = 6.9, 1.2 Hz), 5.26 (t, 1H, *J* = 7.0 Hz). 4.21 (d, 2H, *J* = 6.8 Hz), 2.23–2.19 (m, 4H), 2.14–2.10 (m, 4H), 1.73 (s, 3H), 1.19 (s, 6H), 1.17 (s, 6H). ^13^C NMR (125 MHz, CDCl_3_ + (PhNH)_2_): *δ* = 139.2, 132.3, 125.3, 123.7, 60.3, 60.2, 59.4, 49.5,41.6, 39.8, 25.9 16.3.

*(E)-4-(6-bromo-4-methylhex-4-en-1-ylidene)-2,2,6,6-tetramethylpiperidin-1-yloxy (8)*: To the stirred solution of compound 7 (532 mg, 2.0 mmol) and triethylamine (222 mg, 2.2 mmol) in dry CH_2_Cl_2_ (20 mL), after cooling to −30 °C, methanesulfonyl chloride (249.3 g, 2.2 mmol) was added dropwise. After the mixture temperature increased to room temperature, brine (20 mL) was added, the phases were separated, and the aqueous phase was extracted with CH_2_Cl_2_ (3 × 20 mL). The combined organic phases were dried over MgSO_4_, filtered, and evaporated. The residue was dissolved in anhydrous acetone (30 mL), to which LiBr (522 mg, 6.0 mmol) was then added, and the mixture was stirred at 40 °C for 30 min. Water (20 mL) and Et_2_O (15 mL) were added, and after separation, the water phase was extracted with further Et_2_O (2 × 20 mL). The combined organic phases were dried over MgSO_4_, filtered, and evaporated to achieve paramagnetic geranyl bromide (8) as a reddish-brown oil with a 72% (410 mg) yield. MS *m*/*z* (%) 284/286 (M^+^, 5/2), 248 (6), 109 (13,) 107 (21), 95 (40), 74 (100). NMR measurements cannot be performed without the loss of the bromide function in the presence of hydrazobenzene.

*(E)-4-((6-(1-hydroxy-2,2,6,6-tetramethylpiperidin-4-ylidene)-3-methylhex-2-en-1-yl)oxy)-7H-furo[3,2-g]chromen-7-one (10)*: Bergaptol (9) (202 mg, 1.0 mmol) was dissolved in anhydrous acetone (10 mL), K_2_CO_3_ (276 mg, 2.0 mmol), NaI (19.5 mg, 0.13 mmol), and bromide 8 (427 mg, 1.5 mmol) were added and the mixture was stirred at 40 °C for 24 h under a N_2_ atmosphere. After cooling to room temperature, brine (20 mL) and CH_2_Cl_2_ (20 mL) were added, the organic phase was separated, and the aqueous phase was washed with CH_2_Cl_2_ (2 × 20 mL). Then, the combined organic phase was dried (MgSO_4_), filtered, evaporated, and the residue was purified by flash column chromatography (hexane-EtOAc) to yield the title compound 10 as red crystals, 90 mg (20 %). M.p.: 51–53 °C. R_f_: 0.43 (hexane-EtOAc, 2:1). MS m/z (%) 450 (M^+^, 3), 435 (3), 420 (5), 202 (25), 98 (66), 74 (100) Anal. calcd. for C_27_H_32_NO_5_: C 71.98; H 7.16; N 3.11. Found: C, 72.11; H, 7.20; N, 3.17. IR (neat): $\bar{\nu }$ = 1725, 1623, 1332 cm ^−1^. ^1^H NMR (500 MHz, CDCl_3_ + (PhNH)_2_): *δ* = 8.19 (d, 1H, *J* = 9.8 Hz), 7.60 (d, 1H, *J* = 2.3 Hz), 7.20 (s, 1H), 6.96 (d, 1H, *J* = 2.3), 6.32 (d, 1H, *J* = 9.8 Hz), 5.95 (dt,1H, *J* = 6.8, 1.0 Hz), 5.23 (t, 1H, *J* = 6.8 Hz), 4.97 (d, 2H, *J* = 6.8 Hz), 2.24–2.18 (m, 6H), 2.10 (s, 2H), 1.75 (s, 3H), 1.18 (s, 6H), 1.16 (s, 6H). ^13^C NMR (125 MHz, CDCl_3_ + (PhNH)_2_): *δ* = 161.2, 158.1, 148.9, 144.9, 142.6, 139.5, 132.7, 124.8, 119.2, 114.2, 112.6, 107.5, 105.0, 94.2, 69.7 60.3, 60.2, 49.5, 41.7, 39.8, 25.7, 16.7.

### 4.2. CYP Inhibition Assays

#### 4.2.1. Reagents

Testosterone, 6*β*-hydroxytestosterone, ketoconazole, racemic warfarin, ticlopidine hydrochloride, CypExpress^TM^ 2C9 kit, CypExpress^TM^ 2C19 kit, and CypExpress^TM^ 3A4 kit were purchased from Sigma-Aldrich (St. Louis, MO, USA). Diclofenac, 4′-hydroxydiclofenac, S-mephenytoin, and 4-hydroxymephenytoin were obtained from Carbosynth (Compton, Berkshire, UK).

#### 4.2.2. CYP Assays

Stock solutions of BM, SL-BM, and positive controls (each 5000 μM) were prepared in dimethyl sulfoxide (Fluka) and stored at –20 °C. The inhibitory effects of BM and SL-BM on the CYP2C9, 2C19, and 3A4 enzymes were examined using CypExpress^TM^ assay kits by employing diclofenac, S-mephenytoin, and testosterone substrates, respectively. During these assays, racemic warfarin (2C9), ticlopidine (2C19), and ketoconazole (3A4) were applied as positive controls. The CYP2C9, 2C19, and 3A4 assays were performed as described in our previous studies [39,40,41,42].

The substrates and products were analyzed using the HPLC system comprising a pump (Waters 510; Milford, MA, USA), an injector (Rheodyne 7125) with a 20 μL sample loop, and a UV detector (Waters 486). The data were evaluated by employing Millennium Chromatographic Manager software (Waters). HPLC analyses of diclofenac and 4′-hydroxydiclofenac (CYP2C9 assay) [39,40], S-mephenytoin, and 4-hydroxymephenytoin [42] as well as testosterone and 6*β*-hydroxytestosterone (CYP3A4 assay) [41] were performed as described previously.

Data were derived from at least three independent experiments and represented as the mean ± the standard error of the mean (SEM) values. The statistical significance (*p* < 0.05 and *p* < 0.01) was established by employing a one-way ANOVA test (IBS SPSS Statistics, v. 21; Armonk, NY, USA). The metabolite formation (% of control) was plotted as a function of the logarithmic concentrations, and then the IC_50_ values were determined using GraphPad Prism 8 software (San Diego, CA, USA).

### 4.3. Modeling Studies

#### 4.3.1. Ligand Preparation

The raw ligand structures of ketoconazole, bergamottin (BM, **1**), and SL-BM (**10**) were built in Maestro [43] and energy-minimized with a quantum chemistry program package, MOPAC [33], with PM7 parametrization [44]. Force calculations were also performed using MOPAC, in which the gradient norm was set to 0.001, and the force constant matrices were positive and definite. Gasteiger–Marsilli partial charges were assigned in AutoDock Tools [45]. Flexibility was allowed on the ligand at all active torsions. These prepared structures were used for docking.

#### 4.3.2. Target Preparation

Similar to a previous study [46], the holo structure of CYP3A4 in the complex with ketoconazole was used as a target. The atomic coordinates of the complex were obtained from the Protein Data Bank (PDB) with PDB code 2v0m. The bound ketoconazole molecules were removed prior to the docking calculations, and chain A was processed further as a target. The atomic partial charges of the heme were adopted as the ferric penta coordinate high-spin charge model from reference [47]. The rest of the target molecule was equipped with polar hydrogen atoms and Gasteiger–Marsilli partial charges in AutoDock Tools.

#### 4.3.3. Docking

All the ligand structures were docked to the active site of CYP3A4 using AutoDock 4.2.6 [45]. The number of grid points was set to 90 × 90 × 90 using a 0.375 Å grid spacing. The Lamarckian genetic algorithm was used for the global search, and the flexibility of all active torsions was allowed on the ligand. Ten docking runs were performed, and the resulting ligand conformations were ranked by their free binding energy values. Representative docked ligand conformations were used for the subsequent evaluations, and a collection of interacting target amino acid residues with a 3.5 Å cut-off distance were calculated for heavy atoms. The root-mean-squared deviation (RMSD) values were calculated between the crystallographic and representative ligand conformations.

### 4.4. Cell Viability Assays

Murine embryonic fibroblast NIH3T3 cells were maintained in DMEM (Cat No. 043-30085, Wako Pure Chemical, Osaka, Japan) supplemented with 10% newborn calf serum (NBCS) at 37 °C and 5% CO_2_. Human cervix carcinoma HeLa cells were maintained in DMEM supplemented with 10% fetal bovine serum at 37 °C and 5% CO_2_. To determine the toxicity of BM and SL-BM, the WST-1 cell viability assay was employed as described previously [26]. The cells (NIH3T3: 2000 per well; HeLa: 5000 per well) were seeded into all the wells of 96-well plates. After 24 h, the medium was replaced, and the cells were treated with the test compounds for 24 h. Then, the cells were incubated with a WST-1 solution (3.6 µg/µL WST-1, 70 ng/µL 1-methoxy phenazine methosulfate in 20 mM 4-(2-hydroxyethyl)-1-piperazineethanesufonic acid–KOH (pH 7.4); Dojindo, Kumamoto, Japan) for 1 h at 37 °C, and the absorbance of each well was recorded at 440 nm using a Multiskan FC microplate reader (Thermo Fisher Scientific). The IC_50_ values and statistical significance were calculated using GraphPad Prism 8 software (San Diego, CA, USA).

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
