# Peer review of "Synthesis of Spin-Labelled Bergamottin: A Potent CYP3A4 Inhibitor with Antiproliferative Activity"

_ijms, 2020, doi:10.3390/ijms21020508_

Round 1

Reviewer 1 Report

Article: “Synthesis of spin labelled bergamottin a potent 3 CYP3A4 inhibitor with antiproliferative activity” is well written. In this article, authors gave the synthesis of spin labelled bergamottin (SL-BM), analysis of its inhibition of CYP enzymes compared to bergamottin (BM) and modeling studies.

In my opinion, this paper is acceptable for the publication in “International Journal of Molecular Sciences”.

Author Response

Thank You for Your time and suggestion to accept the manuscript in the present form.

Reviewer 2 Report

This manuscript describes an interesting drug discovery study based on incorporating a redox active spin label into a molecule known to have antiproliferative activity and drug metabolizing enzyme inhibitory activity. The bergamottin side chain is a good choice for a spin label. The synthetic schemes lead to the stated compounds. There is reasonable molecular modeling evidence that supports the similarities of the synthesized analogue Z-isomer with the parent bergamottin for interactions with cyp 3A4, which is indicative of potential drug interaction liability. The nitroxide modification increased the antiproliferative activity in HeLa cells in comparison to that of bergamottin, which is a promising result.

More details of the structure proof of the geometric isomers would be useful. The text mentions a 4:1 ratio of E- to Z-isomers of the test compound, but only the Z-isomer has been synthesized and modeled. It would be interesting to also isolate and model the E-isomer, and to determine the enzyme inhibitory and antiproliferative activities of this isomer.

Because the nitroxide radical is a less common functional group in drug discovery, additional discussion on the properties of nitroxide radicals in drug molecules would be beneficial. In particular, it could be noted that nitroxides are reduced in vivo quite rapidly to hydroxylamine metabolites. Synthesis and study of the analogous hydroxylamine isomers would be a welcomed addition to the synthesis, modeling and biological evaluation aspects of the work.

Author Response

Thank You for Your time and suggestions.

I agree with the comment. We tried the isolation of the neryl derivative, but we could not separate it in pure form by column chromatography. To increase the ratio of Z ester we tried Still-Gennari modification of Horner-Wadsworth-Emmons reaction, but our ketone was not reactive below – 30oC. The best achieved ratio of neryl/geranyl esters was 1:3. The 4:1 ratio was estimated from row NMR, containing both isomers, where the triplet of vinylic H with a chemical shift of 5.26 had a triplet pair with a shift of 5.33 (peak area were 0.8 vs. 0.2) and C13 NMR gave similar duplications at C=C carbons.

Nitroxide-N-hydroxylamine conversion is especially fast in the case of piperidine type of nitroxides. After reaching the equilibria the normal cells contain about 30% N-O and 70% the N-OH, hypoxygenic cancer cells about 10% N-O and 90 % N-OH respectively. The investigation of diamagnetic (N-OH) bergamottin derivative can be interesting pharmacologically, nevertheless our main goal was the synthesis of paramagnetic analog of bergamottin. Please accept our answers.

The English of the manuscript was checked by ACS authoring service, we hope it will be fine now.